# Interpretable Prediction of Lung Squamous Cell Carcinoma Recurrence With Self-supervised Learning

**Weicheng Zhu**[1]                   JACKZHU@NYU.EDU
**Carlos Fernandez-Granda**[*1,2]          CFGRANDA@CIMS.NYU.EDU
**Narges Razavian**[*3]        NARGES.RAZAVIAN@NYULANGONE.ORG

[1] *Center for Data Science, NYU*

[2] *Courant Institute of Mathematical Science, NYU*

[3] *Departments of Population Health and Radiology, NYU Grossman School of Medicine*

## Abstract

Lung squamous cell carcinoma (LSCC) has a high recurrence and metastasis rate. Factors influencing recurrence and metastasis are currently unknown and there are no distinct histopathological features indicating the risks of recurrence and metastasis in LSCC. Our study focuses on the recurrence prediction of LSCC based on H&E-stained histopathological whole-slide images (WSI). Due to the small size of LSCC cohorts in terms of patients with available recurrence information, standard end-to-end learning with various convolutional neural networks for this task tends to overfit. Also, the predictions made by these models are hard to interpret. In this work, we propose a novel conditional self-supervised learning (SSL) method to learn representations of WSI at the tile level first, and leverage clustering algorithms to identify the tiles with similar histopathological representations. The resulting representations and clusters from self-supervision are used as features of a survival model for recurrence prediction at the patient level. Using two publicly available datasets from TCGA and CPTAC, we show that our LSCC recurrence prediction survival model outperforms both LSCC pathological stage-based approach and machine learning baselines such as multiple instance learning. The proposed method also enables us to explain the recurrence risk factors via the derived clusters. This can help pathologists derive new hypotheses regarding morphological features associated with LSCC recurrence. Code available at https://github.com/NYUMedML/conditional_ssl_hist.

**Keywords:** self-supervised learning; clustering; survival analysis; histopathology

## 1. Introduction

Histopathological features are useful in identification of tumor cells, cancer subtypes, and the stage and level of differentiation of the cancer. Hematoxylin and Eosin (H&E)-stained slides are the most common type of histopathology data and the basis for decision making in the clinics. H&E-stained slides include several cellular morphological features but the relationship between these features and patient prognosis or genetic mutation of the corresponding tumor tissue remains unknown. Current findings are primarily based on pathologist expertise. For example, the papillary pattern in lung adenocarcinoma is a recognizable signal of invasive tumor cells and poor prognosis (high recurrence rate). However, these predictive

---

* Contributed equally

morphological patterns are not available for many other cancer subtypes. Specifically, at the moment, there are no known pathological patterns associated with recurrence in lung squamous cell carcinoma (LSCC).

Deep learning has achieved promising results in several classification and prediction tasks based on histopathology images (Bejnordi et al., 2017; Coudray et al., 2018; Wulczyn et al., 2020; Fu et al., 2020; Kather et al., 2020). In this paper, we focus on predicting LSCC recurrence and metastasis using H&E stained slides. Each whole slide images (WSI) contains more than $3.5 \times 10^8$ pixels on average, and can be cropped into hundreds of tiles, but the label is at the slide/patient level. A typical cohort with high quality data, at the moment, only has around 500 patients. While for several tasks such as cancer subtype detection or cancer vs normal cell identification, such cohorts have led to successful models (Bejnordi et al., 2017; Coudray et al., 2018; Campanella et al., 2019; Iizuka et al., 2020; Hong et al., 2021), prediction of other outcomes such as recurrence or genomics mutation remain challenging using standard supervised learning (Fu et al., 2020; Kather et al., 2020; Wulczyn et al., 2021). Most prior work extends slide-level labels to all the tiles within the slide, which is a reasonable assumption in subtype prediction tasks, but is not valid in other classification tasks. Lack of tile-level labels remains one of the challenges in predicting cancer recurrence.

Self-supervised learning (SSL), which leverages unlabeled images to learn model parameters, is an alternative approach that can utilize the rich tile-level image data. Once histopathological features are trained, they can be fine-tuned for various downstream classification tasks using fewer labeled data (Li et al., 2020). Also, the pretrained image representations can also be used to interpret histopathological features. Wulczyn et al. (2021) interprets the histopathological features associated with survival of colon-cancer patients by clustering tile-level embeddings pretrained with natural images. Models trained via SSL on histopathological data can learn domain-specific features and are better suited for downstream tasks compared to natural images. However, naive SSL application can also be susceptible to *batch effects*, which are a common problem in histopathology. Models trained via standard SSL methods are likely to learn undesirable features due to the batch effect, which may lead to overfitting (Tsai et al., 2021b).

In this study, we explore the morphological features of LSCC recurrence and metastasis with novel SSL method, based on conditional SSL. We propose a sampling mechanism within contrastive SSL framework for histopathology images that avoids overfitting to batch effects. Using our proposed SSL training combined with clustering, we show significant improvement in the recurrence prediction on the LSCC compared to pathological stage-based model, and several deep learning baselines. We also provide interpretation of the learned model to identify morphological patterns that may be associated with LSCC recurrence.

## 2. Related Work

**Representation Learning for Histopathology** Whole Slide Images (WSI) are large and are therefore usually cropped into several tiles, and the tiles are used as units of data. Ground-truth annotations for each tile are not available in many tasks, because this would require intensive human labor, or is infeasible. For instance, for some tasks such as prognosis, only patient-level labels are available. A prevalent approach in the literature has been to train a tile-level network with slide-level label using standard supervised learning, and

pool the tile-level predictions during inference (Wang et al., 2016; Coudray et al., 2018; Fu et al., 2020; Iizuka et al., 2020; Hong et al., 2021; Bejnordi et al., 2017; Campanella et al., 2019; Kather et al., 2020). Another approach is multiple instance learning (MIL) (Ilse et al., 2018), which interprets each slide as a bag of instances, each corresponding to a tile. However, computer memory is a significant constraint when training these models. Leveraging pretrained convolutional neural network (CNN) weights is a shortcut to avoid this memory issue. Pretraining can be carried out via fully-supervised tasks like tumor classification (Fu et al., 2020), ranking tasks on nature images (Hegde et al., 2019), MIL with downsampled bags (Zhao et al., 2020), or self-supervised learning.

**Self-supervised Learning**   Recent advances in self-supervised learning (SSL) for computer vision have improved the quality of latent representations in cases without sufficient annotated samples. These methods have shown promising results in medical imaging (Li et al., 2020; Ciga et al., 2022; Dehaene et al., 2020; Sowrirajan et al., 2020; Azizi et al., 2021; Kaku et al., 2021). Contrastive SSL is currently the state-of-the-art SSL method for natural images (Chen et al., 2020b,c; Tsai et al., 2021a; Chen et al., 2020a; He et al., 2020; Grill et al., 2020; Caron et al., 2020; Zbontar et al., 2021; Caron et al., 2021). In this approach, models are trained to increase similarity of representations corresponding to augmented copies of the same image, while decreasing the similarity of augmented copies from different images. Contrastive SSL has been applied to histopathology (Li et al., 2020; Ciga et al., 2022). However, the contrastive loss, InfoNCE (van den Oord et al., 2018) is not tailored to the special properties of WSIs. Robinson et al. (2021) show that contrastive loss tends to learn "shortcuts" within the similar and dissimilar samples, which inadvertently suppress important predictive features. In histopathology, features associated to slide-specific batch effects (i.e. staining, procedural artifacts, etc) are captured by contrastive learning more easily than meaningful pathological features, as we show in Section 3. Some methods have been proposed to improve SSL for multiple instance learning approach. Azizi et al. (2021) use instances from the same bag as positive augmented samples instead of general augmentation. However, the method promotes similarity between tiles from the same slide, which makes it even harder to avoid batch effects. Conditional contrastive SSL (Tsai et al., 2021b) has been proposed to remove the undesired features in contrastive learning by conditionally sampling tiles based on these features during the training. It can potentially remove undesired batch-effect-induced feature such as slide identity in the histopathological case, but as we will show in Section 5 standard conditional SSL is also sub-optimal in learning global features differentiating histopathological variations between different slides.

**Survival Analysis**   The likelihood of LSCC recurrence varies as the time progresses, so we use survival analysis tools to model the risk of LSCC recurrence over time. The Kaplan-Meier estimator (Kaplan and Meier, 1958) is a non-parametric model that estimates the survival function over time. It shows what fraction of patients remain recurrence-free for a certain amount of time after treatment. Cox proportional hazard regression (Cox, 1972) is a parametric method to model the effect of variables on the time a specified event is expected to happen. Variants of Cox regression method have been proposed to extract features from complex or high-dimensional data such as images. Cox-nnet (Ching et al., 2018) and DeepSurv (Katzman et al., 2018) perform regression with multiple-layer perceptron (MLP) networks. They are also compatible with other feature extraction networks such as

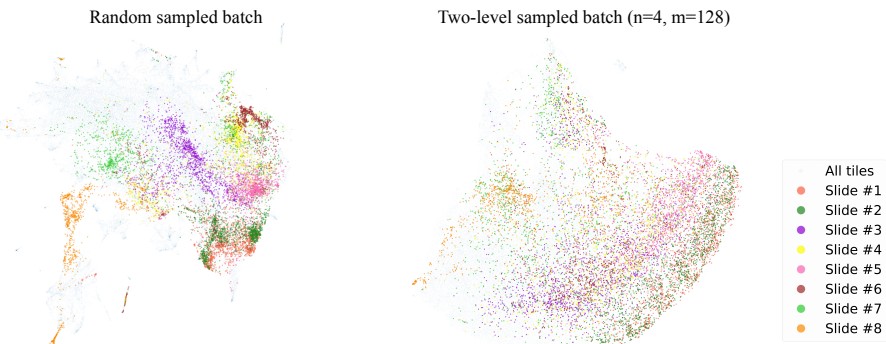

Figure 1: The 2D UMAP projection of tile representations, trained by different sampling in self-supervised learning. Tiles from 8 slides with mostly LSCC tumor content are highlighted with different colors. **Left:** model trained by MoCo contrastive learning with uniform sampling. It shows that tiles within each slide cluster together. **Right:** model trained with proposed conditional contrastive learning. The tiles from each slide are less clustered together.

CNNs. Another approach to survival analysis is to transform the regression into multiple classification problems over discretized time periods (Gensheimer and Narasimhan, 2019).

## 3. Proposed Method

**Conditional Contrastive Learning** Most contrastive SSL methods including SimCLR (Chen et al., 2020a) and MoCo (He et al., 2020) optimize InfoNCE

$$\mathcal{L}(x, x^+, \{x^-\}) = -\log \frac{\exp\left(g(x, x^+)\right)}{\exp(g(x, x^+)) + \sum_{x^-} \exp\left(g(x, x^-)\right)} \tag{1}$$

where $x, x^+, \{x^-\}$ are the output embeddings of the image encoder model of an augmented image, another random augmentation of the same image, and augmented versions of different images within a batch, respectively. In our case, we use Inception-v4 (Szegedy et al., 2016) as the encoder, and augmentations as listed in Appendix A.1. $g(x, x')$ is a function computing the similarity between $x$ and $x'$. Previous studies on histopathology with contrastive learning, Lu et al. (2019); Dehaene et al. (2020); Li et al. (2020) train the network with tiles from WSIs. The source images of $x^+$ and $\{x^-\}$'s are sampled randomly with equal probabilities among all the tiles. However, when the batch size is not much greater than the number of slides, there hardly are tiles from the same slide in a batch. Hence, The probability of forming negative pairs from tiles in the same slide is low. The network tends to learn features caused by slide-level batch effects, which are easily captured by contrastive learning. Figure 1 (left panel) shows the UMAP projection of the representations of tiles learned using standard contrastive SSL (MoCo (He et al., 2020)) with random sampling. We can see that tiles from the same slide tend to cluster together, indicating that this method learns representations that are slide-specific and may lead to low generalization when presented with new patients.

As an alternative, conditional contrastive learning (Robinson et al., 2021) optimizes *C-InfoNCE*, where $x^+, \{x^-\}$ are sampled given some condition $z$ (i.e. having the same slide id) in histopathology. However, as we will show in Section 5, this approach also prevents the model from learning features that differentiate between slides and are also useful for the downstream task of interest. In order to learn from both inter-slide and inter-tile variations, we propose a two-level sampling method during training. Suppose the batch size is $m$, we first randomly sample $n$ slides ($n \ll m$), and then sample $m/n$ tiles from each selected slide. Figure 1 (right panel) shows the UMAP projection of the representation of tiles leaned using the two-stage sampling approach. Tiles from the same slide no longer cluster together, indicating that slide-specific features are less significant using this SSL approach. Consequently, conditional SSL is able to achieve better performance in the downstream task than the classic MoCo, as we will show in Section 5.

**Clustering the Representation Space**  Unsupervised clustering algorithms have been shown to be an effective tool for processing SSL representations (Caron et al., 2018). We use a Gaussian Mixture Model (GMM) (Pedregosa et al., 2011) to cluster the tile representations in the training set. GMM is a probabilistic model that estimates the probability of a sample belonging to a cluster. GMM assumes that samples $\{x_i\}_{i=1}^N$ are generated from a mixture of $k$ Gaussian distributions. It computes the probability that each tile embedding $x_i$ belongs to each $z$ among the $k$ clusters (i.e. $p(z \mid x_i)$). After clustering each tile, we aggregate tile probabilities to generate the slide-level features. Suppose the slide $S_j$ is composed of tiles whose representations are $x_s \in S_j$. The new slide-level feature $v_j \in \mathbb{R}^k$ is assigned by the average pooling of probabilities over the slide. i.e. $v_j[z] = \frac{1}{|S_j|} \sum_{x_s \in S_j} p(z \mid x_s)$, where $v_j[z]$ is the $z$-th entry of $v_j$, for $z = 1 \cdots k$. Analyzing the clusters can help with interpretability, revealing common morphological patterns in the cells that may be associated with cancer recurrence. Therefore, we use the cluster-generated features in our prediction model.

**Survival modeling**  In order to predict recurrence, we combine the cluster features obtained via GMM with a survival-analysis model. For each slide, the triplet of features and slide labels $\{(v_j, y_j, t_j)\}_{j=1}^N$ will be used, where $v_j$ is the vector of cluster features, $y_j$ is the binary label indicating LSCC recurrence, and $t_j$ encodes the recurrence-free followup times for the patient. i.e. If a patient was not observed to have recurrence during the followup period, we use the length of followup time $t_j$ as the time of censoring. Each $t_j$ is computed with a granularity of 6 months. We fit a Cox regression model with $L_2$-norm regularization using $\{(v_j, y_j, t_j)\}_{j=1}^N$ to compute the proportional hazard function of recurrence $\lambda(t|v)$.

## 4. Experiments

**Data**  This study analyzed lung squamous cell carcinoma (LSCC) patient data, including hematoxylin-and-eosin stained (H&E) histopathology slides from frozen specimens, recurrence status, and demographic information, from two cohorts - The Cancer Genome Atlas Program (TCGA) and Clinical Proteomic Tumor Analysis Consortium (CPTAC). TCGA and CPTAC datasets have 824 and 511 slides from 504 and 215 patients with LSCC, respectively. The SSL training was performed using all cancerous slides of LSCC patients except those used in downstream task validation and test sets. For the downstream task, we excluded patients with new primary tumor (11 in TCGA and 5 in CPTAC). In the remaining

| Method | | Metrics | |
|---|---|---|---|
| Feature | Survival model | C-index | Brier score (2 years) |
| Stage (*) | KM estimator | $0.548 \pm 0.053$ | $0.202 \pm 0.050$ |
| Tile images (E2E) | DeepSurv | $0.529 \pm 0.048$ | $0.217 \pm 0.038$ |
| | NN-Surv | $0.534 \pm 0.010$ | $0.211 \pm 0.034$ |
| Tile bags (MIL) | DeepSurv | $0.589 \pm 0.044$ | $0.203 \pm 0.030$ |
| | NN-Surv | $0.595 \pm 0.018$ | $0.207 \pm 0.029$ |
| SSL clusters | Cox Reg. | $\mathbf{0.646 \pm 0.055}$ | $\mathbf{0.200 \pm 0.051}$ |

Table 1: The performance of recurrence prediction (0.95-confidence intervals are computed using the standard deviation of 5 trials). (*) Stage is extra information that is not used in the other methods.

cohort, 57 LSCC patients in TCGA, and 29 LSCC patients in CPTAC suffered from recurrence or metastasis during the followup time after surgery. Details for preprocessing the slides are described in Appendix A.2.

We combine the data from two datset and split them by patient and institution (TCGA contains data from 42 institutions) into train, validation, and test sets containing 70%, 10%, and 20%, respectively. Since the size of dataset is small, analysis was conducted by nested cross validation, replicating training and testing procedures in five different splits.

**Baseline Models** Our baselines include a pathological stage-based model and a number of deep learning methods. (details on models and inference are in Appendix B).

***Pathological stage*** Pathologists evaluate the progress of cancer with stages. Patients at higher stages are more likely to suffer from recurrence. It is a clinical "golden rule" to estimate recurrence and metastasis prognosis. In the case of LSCC, at the moment, it is the only method available for estimating the prognosis. We use Kaplan–Meier estimator to compute the empirical recurrence hazard of each stage over time.

***Deep survival models*** We use the continuous-time DeepSurv (Katzman et al., 2018) and the discrete-time model NN-Surv (Gensheimer and Narasimhan, 2019) as baselines.

## 5. Results

We evaluate the performance of recurrence prediction using the concordance index (C-index) and Brier score at 2-year, comparing our approach to the baselines in Section 4. C-index equals to the ratio of concordance between recurrence-free time and predicted risk. C-index measures the **discriminative** power of a survival model. Brier score at time $t$ is the mean square error between recurrence status and predicted probability on recurrence at time $t$, weighted by the inverse probability of censoring. The Brier score is a metric for **calibration and probability estimation** (Liu et al., 2021).

In Table 1, we report the performance of recurrence prediction. The tumor's pathological stage provides a biologically motivated baseline, and the C-index of stage shows that predicting LSCC recurrence based on this metric remains a hard task. Our Cox regression

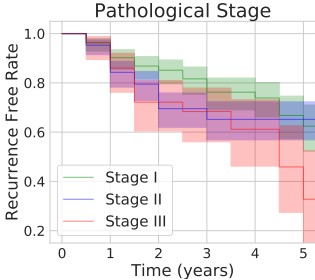 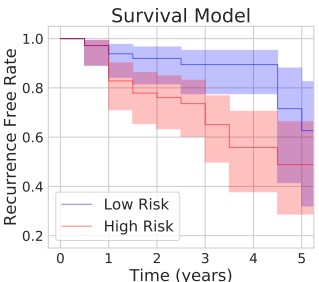

Figure 2: The Kaplan-Meier curves shows rates of recurrence-free patients over time in sub-cohorts of test set with different criterion. **Left:** three sub-cohorts stratified with pathological stages; **Right:** two sub-cohorts stratified with the predicted recurrence risk by our Cox regression. The high risk cohort includes the top half patients of highest estimated risks; the low risk cohort includes the lower half.

model based on SSL clusters improves the C-index by 10% , which is a significant improvement beyond the current clinical approach. The Kaplan-Meier (KM) curves in Figure 2 also show the gap between the machine learning and stage prediction. The left panel shows that Stage I has slightly less recurrence rate than Stage II and III. However, three curves have significant overlap. The right panel in Figure 2 shows the KM curves for cancer recurrence on the heldout test set according to our model. High risk (and low risk) are defined as top half (and bottom half) of patients according to recurrence risks, respectively. The results show that high vs. low risk patients can be differentiated with our method.

We also compared our method with other deep survival baseline models in Section 4. All of the baselines under-perform our SSL-based clustering approach. Among these baselines, we find the models with multiple instance learning (MIL) to generally perform better than end-to-end (E2E) learning with tile images and slide-level labels. MIL can avoid the inconsistency between the tile and slide labels. Also, MIL takes advantage of pretrained CNN with frozen weights, which may alleviate the overfitting. Comparing two type of survival models, the differences between discrete and continuous-time models are not significant. We also quantitatively evaluated the impact of the batch effect on contrastive SSL, illustrated in Figure 1. We experimented with different combinations of sampling in conditional contrastive learning, and evaluated their performance on the downstream recurrence prediction,

| $(n)$ | 1 (C-InfoNCE) | 4 | 16 | 32 | *Random* |
|---|---|---|---|---|---|
| C-index | $0.631 \pm 0.052$ | $\mathbf{0.646 \pm 0.055}$ | $0.624 \pm 0.037$ | $0.599 \pm 0.109$ | $0.601 \pm 0.069$ |
| Brier (2 yrs) | $0.225 \pm 0.046$ | $\mathbf{0.200 \pm 0.051}$ | $0.227 \pm 0.044$ | $0.224 \pm 0.049$ | $0.202 \pm 0.049$ |

Table 2: The performance of recurrence prediction with conditional SSL by sampling different number $(n)$ of slides in each batch when batch size $m = 128$. *Random* uses the classic MoCo with random sampling. The model with $n = 1$ uses C-infoNCE.

Large clusters of tumor (40)   Tumor with infiltration (48)   Ring structured tumor (43)

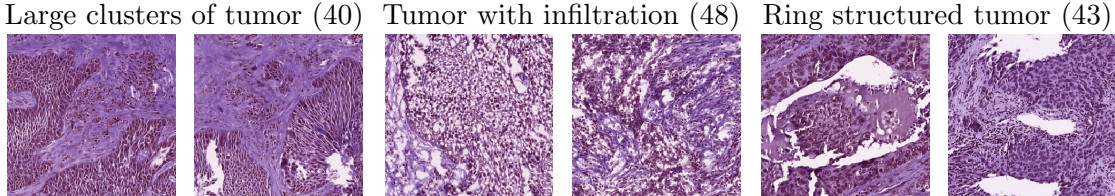

Figure 3: Visual samples from clusters with positive regression coefficients in our survival model for recurrence prediction (more samples in Figure 5 Appendix D)

shown in Table 2. We kept the batch size the same in terms of number of tiles, and varied the number of slides in each batch in the first sampling layer, which ranges from 1 (the fully conditional case) to random sample (the classic MoCo). The model performance peaks at $n = 4$. As $n$ increases, the models are more likely to learn features corresponding to batch effects. When $n = 32$, the C-index becomes close to random sampling. The standard C-InfoNCE ($n = 1$) prevents the contrastive learning method from learning variations among different slides beyond the batch effect. The experiments show that two-layer sampling with the appropriate $n$ can achieve a good balance between the two extremes.

**Discussion**   To interpret the histopathological features indicating a high risk of recurrence, we evaluate the association between each tile cluster and recurrence based on our survival model. As the hazard proportion are computed by $\lambda(t|v) = \lambda_0(t) \exp(\beta^T v)$, we apply an exponential function to the Cox regression coefficients corresponding to the cluster features in order to obtain a measure of feature importance, reported in Appendix C. If the result is greater than one, then the corresponding cluster is positively associated with recurrence.

Figure 3 shows the description on the dominant morphological pattern visible in the top clusters associated with high risk of recurrence in the Cox regression. We select tiles with highest probability belonging to each cluster. The selected clusters and tiles were reviewed by a pathologist who summarized the dominant pathological features in the cluster. High risk histopathological features include large tumor clusters, tumors with infiltration, and ring structure of tumor cells. This may yield novel hypothesis to explore the causes of LSCC recurrence. A number of small outlier clusters also occur in our analysis, like cluster 49, that indicate frozen or slicing artifacts occasionally occuring in different parts of WSIs. Future studies can potentially exclude these tiles, as they are irrelevant to biological features.

## 6. Conclusion and Future Work

In this paper, we leveraged conditional self-supervised contrastive learning to learn the morphology of whole slide images, and analyze the features associated with lung squamous cell cancer recurrence and metastasis. To refine the self-supervision in histpathology domain, we proposed a two-layer sampling method to alleviate overfitting slide-level batch effects while retaining strong discriminative performance. Our method outperforms clinical and deep learning baselines for LSCC recurrence prediction. In addition, it makes it possible to identify tissue morphology patterns that may be helpful in identifying future recurrence.

## Acknowledgments

The results published here are in whole or part based upon data generated by the TCGA Research Network: https://www.cancer.gov/tcga and the National Cancer Institute Clinical Proteomic Tumor Analysis Consortium (CPTAC).

The authors gratefully acknowledge support from NRT-1922658 (W.Z.), DMS2009752 (C.F.G.), and the NYU Langone Health Predictive Analytics Unit (N.R., W.Z.). The authors would like to thank the help from Dr. Andre Moreira and Dr. David Fenyo.

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

# Appendix A. Experiment Details

## A.1. Data Augmentation

We used following augmentation for contrastive self-supervised learning:

- Random color jittering with brightness, contrast, and saturation factor chosen uniformly from [0.6,1.4], and hue factor chosen uniformly from [-0.1,0.1], with 0.8 probability;

- Random gray scale with 0.2 probability;

- Random Gaussian blur with radius chosen uniformly from [0.1, 0.2] with 0.5 probability;

- Random horizontal/vertical flipping.

## A.2. Slide Preprocessing

To preprocess the slides, we cropped the slides into tiles with dimension $1024 \times 1024$ at 10x magnification with 25% overlap, and filtered out the tiles with more than 85% area covered by the background (following (Coudray et al., 2018)), and resized the images into $299 \times 299$ pixels. Resizing was performed using an anti-aliasing filter of Pillow package (Clark, 2015). These tiles were color normalized to the stain level of a standard reference image (included in the open source package) using the Vahadane method (Vahadane et al., 2016). The reference image for this method is a LSCC tile image that has clear H&E stain and cell structure to avoid color difference among different stains. To only analyze the cancerous parts of WSIs, all the tiles in the tumor slides without any tumor cells were excluded based on a separate Inception-v4 trained with full supervision to distinguish tumor and normal cells (AUC at 98%)(Coudray et al., 2018).

## A.3. Hyperparameters

For MoCo we use embedding dimension at 128, batch size at 128, temperature scaling at 0.07, and encoder momentum at 0.999. We pretrain the MoCo for 200 epochs.

We tune the hyperparameter $k \in \{10, 50, 100\}$ with both k-means and GMM cluster methods based on the C-index on the validation set. GMM with $k = 50$ has the highest validation C-index. We observe that enlarging the number of clusters will introduce some sparse clusters with few tiles, which makes our cluster-based feature noisier. We also tune the weight of $L_2$-regularization $\alpha$ the Cox proportional hazard model in $[10^{-4}, 10^2]$. We achieve the best validation loss at $\alpha = 0.1$.

# Appendix B. Baselines

**Demographic Analysis** In addition to stage-based model, we also train Cox proportional hazard regression with stage and other demographic features (age and gender). The resulting test C-index is $0.526 \pm 0.023$. It is not improved beyond the prediction only based on stage. Figure 4 shows that there are no significant differences in age and gender distributional over recurrence status.

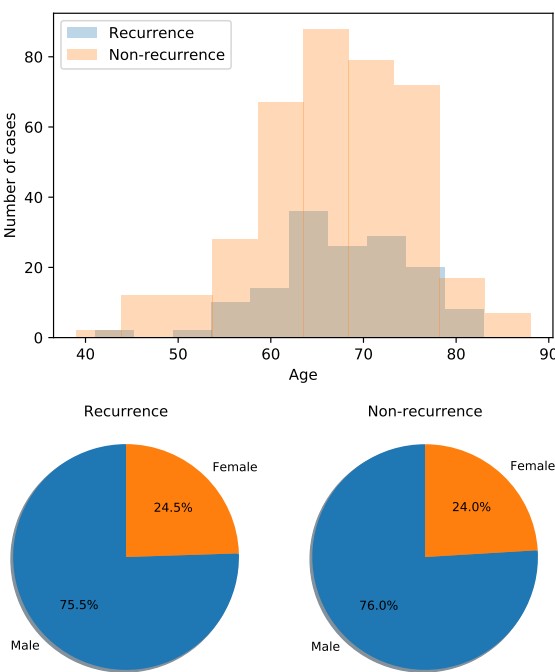

Figure 4: The distribution of age and gender over recurrent and non-recurrent patient

**Survival Models** We experiment with other deep survival models which directly use the WSI tiles as the inputs. We use a continuous-time model (DeepSurv (Katzman et al., 2018)) and a discrete-time model (NN-Surv (Gensheimer and Narasimhan, 2019)) as baselines. DeepSurv, similar to Cox proportional hazard regression, models $\lambda(t|x) = \lambda_0(t) \exp(f(x))$, where $f(x) \in \mathbb{R}$ is network output of sample $x$, and $\lambda_0(t)$ is a baseline proportional hazard function, estimated by Breslow's method (Breslow, 1975). NN-surv models the probability of recurrence separately on each time interval. Assume there are $T$ intervals ($0 \le t_0 < t_1 < \cdots < t_T$), the final layer of NN-surv has $T$ outputs, and each of them represents the probability of recurrence not occurring during that time interval conditioning on the previous interval (i.e. $p(t > t_{i+1} \mid t > t_i)$. The hazard ratio is defined as $\lambda(t|x) = \prod_{i=1}^{j} p(t > t_{i+1} \mid t > t_i, x)), \forall t \in [t_j, t_{j+1})$. We compare our methods with these two types of deep survival models in Section 5.

We apply each model using fully-supervised end-to-end learning, and multiple-instance learning. For end-to-end learning, we conduct the tile-level training with patient-level labels, and aggregate the prediction by averaging over the tiles in each slide at inference. For multiple-instance learning, we take tile representations pretrained by MoCo (He et al., 2020), and learn attention weights over the tile representations to generate a slide-level prediction (Ilse et al., 2018).

# Appendix C. Cox Regression Coefficents

Table 3: The coefficients of Cox regression

| Cluster | $\exp(\beta_i)$ | Cluster | $\exp(\beta_i)$ |
|---|---|---|---|
| Cluster 2 | 6.348966 | Cluster 47 | 0.968074 |
| Cluster 40 | 5.107653 | Cluster 5 | 0.948444 |
| Cluster 49 | 4.921752 | Cluster 4 | 0.948252 |
| Cluster 48 | 4.877566 | Cluster 3 | 0.830124 |
| Cluster 11 | 4.832815 | Cluster 32 | 0.689943 |
| Cluster 15 | 3.254782 | Cluster 33 | 0.653334 |
| Cluster 34 | 2.736110 | Cluster 46 | 0.641871 |
| Cluster 23 | 2.714640 | Cluster 45 | 0.635891 |
| Cluster 43 | 2.670792 | Cluster 18 | 0.613605 |
| Cluster 16 | 2.429464 | Cluster 13 | 0.575792 |
| Cluster 0 | 2.337257 | Cluster 1 | 0.548997 |
| Cluster 14 | 1.967195 | Cluster 28 | 0.543744 |
| Cluster 41 | 1.768265 | Cluster 22 | 0.532754 |
| Cluster 35 | 1.755566 | Cluster 12 | 0.516309 |
| Cluster 31 | 1.729773 | Cluster 17 | 0.463136 |
| Cluster 7 | 1.716123 | Cluster 37 | 0.436561 |
| Cluster 24 | 1.497294 | Cluster 6 | 0.415598 |
| Cluster 26 | 1.493279 | Cluster 10 | 0.375982 |
| Cluster 21 | 1.384650 | Cluster 42 | 0.368304 |
| Cluster 19 | 1.336791 | Cluster 36 | 0.355574 |
| Cluster 39 | 1.168879 | Cluster 25 | 0.308865 |
| Cluster 20 | 1.090216 | Cluster 8 | 0.293442 |
| Cluster 44 | 1.032839 | Cluster 27 | 0.250068 |
| Cluster 30 | 1.022530 | Cluster 9 | 0.141011 |
| Cluster 38 | 1.000819 | Cluster 29 | 0.140645 |

## Appendix D. Cluster Visualization

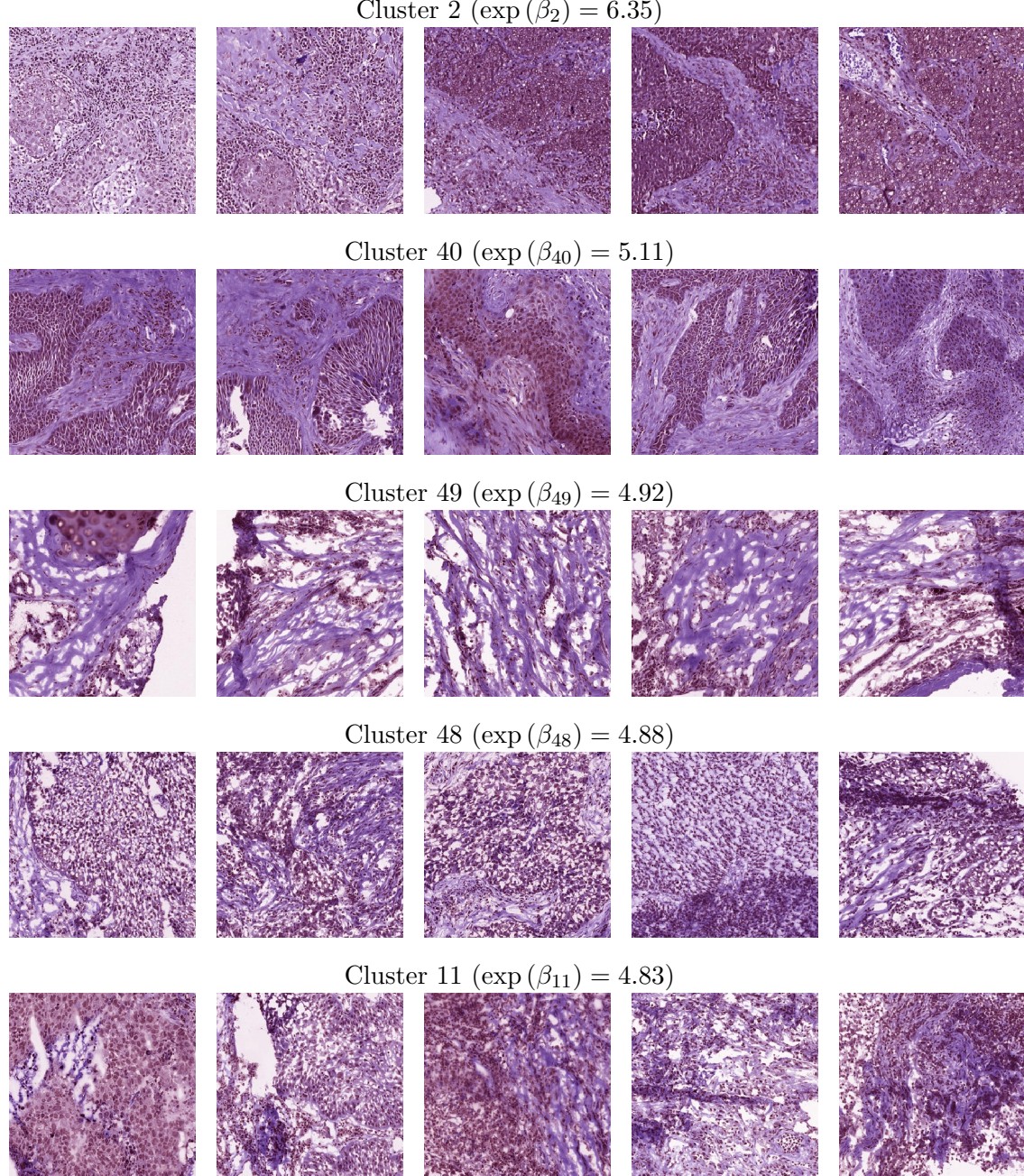

Figure 5: Additional visualization of high risk clusters supplementing Figure 3

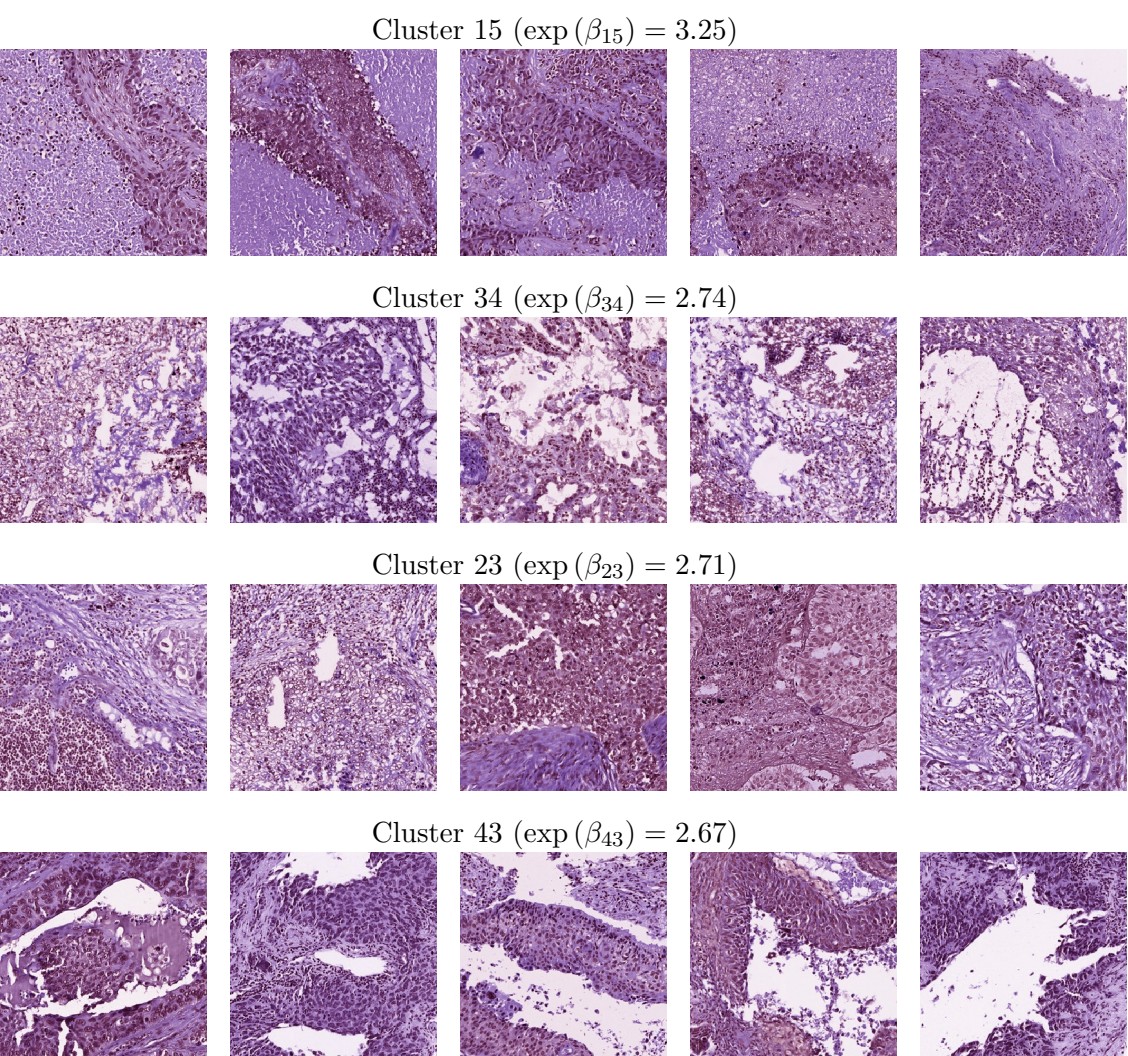

Figure 6: Additional visualization of high risk clusters supplementing Figure 3

