# OpenReview forum: "Interpretable Prediction of Lung Squamous Cell Carcinoma Recurrence With Self-supervised Learning"
_MIDL.io/2022/Conference — MIDL 2022_

### Official Review · Reviewer_gonu · 2022-01-22

**Confidence:** 4
**Preliminary Rating:** 4
**Recommendation:** Poster

**Summary:**

The paper proposes a SSL method for Lung Squamous Cell Carcinoma (LSCC) prediction.
The main contribution of the paper is a sampling mechanism to select positive and negative samples for contrastive learning such that the modal can learn both tile and slide level representations.
Once the representations are learned via contrastive learning with the proposed sampling mechanism, the representation space is clustered using GMM.
Then, cluster features and recurrence follow-up times information are used to fit a Cox regression model which is used for recurrence prediction.
Experiments are performed using 2 datasets and the results are compared with Multiple Instance Learning and end-to-end (I assume fully supervised) learning methods.
The paper also presents a discussion about the interpretability of tile clusters for recurrence prediction which may be useful to come up with novel hypotheses to understand the causes of LSCC.

**Strengths:**

- The proposed sampling mechanism is simple yet effective to learn unsupervised representations in histopathological whole-slide images.
- Interpretability of tile clusters is quite interesting and useful when such approaches are used in practice.
- The paper presents a nice story of what are the missing parts/shortcomings in the previous works and how the proposed method addresses these shortcomings.

**Weaknesses:**

- One of the weaknesses of the paper is the lack of comparison with the other SSL methods in the literature that are successfully applied to different problems. For example, it would be interesting to compare with pretext-task-based methods [1, 2, 3], augmentation-based methods [4, 5], and self-training [6] to position the proposed sampling mechanism in the broad literature for learning from limited labeled data.
[1] Gidaris et al. Unsupervised representation learning by predicting image rotations.
[2] Pathak et al. Context encoders: Feature learning by inpainting.
[3] Chen et al. Self-supervised learning for medical image analysis using image context restoration.
[4] Chaitanya et al. Semi-supervised task-driven data augmentation for medical image segmentation.
[5] Zhang et al. mixup: Beyond empirical risk minimization.
[6] Bai et al. Semi-supervised learning for network-based cardiac mr image segmentation.

- Some hyperparameters are not discussed properly. For example, how k in GMM is chosen and tile size are crucial parameters that would be useful to discuss more.




**Deanonymize Review:**

no

**Detailed Comments:**

- The part where labeled data are used for downstream task is not very clear to me and I think it can be explained better. The usual procedure in SSL is to pre-train an encoder on unlabeled data and train a decoder or MLP for a downstream task on limited number of labeled data. Here, there is no 2nd stage of training part is not very clear. I assume the limited amount of labeled data are used to train the Cox regression model, but I don't see that it is mentioned explicitly.

- The paper uses 2 different datasets in the experiments, however, as far as I understand, these datasets are combined and used as a single dataset. Because there are no separate results for TCGA and CPTAG. Were unlabeled images from both datasets combined and used to train a single network with contrastive learning? Please clarify.

- There is no conference/journal info in some of the references.

**Final Rating After The Rebuttal:**

5: Strong Accept

**Justification Of The Final Rating:**

I would like to thank the authors for the rebuttal addressing the concerns of the reviewers. The proposed sampling strategy for contrastive learning is quite interesting and the paper show sufficient experiments to show the superiority of the method. Also, the rebuttal carefully addresses the concerns raised by the reviewers. Therefore, I update my rating as "strong accept".

**Paper Type:**

methodological development

**Questions To Address In The Rebuttal:**

- What are the effects of hyperparameters I mentioned in the weaknesses section in the proposed method?
- How would the proposed method perform compared to some of the benchmarks listed above for learning from limited labeled data?

**Special Issue:**

no

---

### Official Review · Reviewer_buAU · 2022-01-24

**Confidence:** 4
**Preliminary Rating:** 3
**Recommendation:** Poster

**Summary:**

The authors propose to use a self-supervised learning method to learn representations of tiles of whole-slide histopathology images and later cluster similar representations. Next, they use the representation and cluster as features for a survival model for LSCC (Lung Squamous Cell Carcinoma) recurrence prediction. They evaluated this on two public histopathology datasets and observed improved results compared to pathological stage-based methods and machine learning approaches such as multiple instance learning. They also state that these clusters from representations can help in explaining the recurrence risk factors using the obtained clusters.

**Strengths:**

1. It is a well-written article with good motivation for the addressed problem. It is easy to follow.
2. The authors address an important problem of LSCC recurrence prediction.
3. The experiments are well done, and improved results are observed with the proposed method over the compared baselines.


**Weaknesses:**

P1. The authors claim that the modified sampling strategy to learn inter-slide and inter-tile representations. How are the positive and negative samples chosen for the conditional InfoNCE loss to ensure these inter-slide and inter-tile differences are learned during training. This information is not clear from the current text. Can the authors please provide more details on this selection of positive and negative pairs to achieve this based on the m/n tiles from each chosen slide (n slides sampled randomly)?

P2. The authors claim that using InfoNCE loss directly in this application can lead to learning not so useful representations and will contain batch effects. How does the proposed way of sampling avoid this batch effect? Can the authors please provide more details to clarify this?

P3.The inter-slide difference for the representations is not clear from figure 1 (right panel). Can the authors please provide more text describing what to look for in the figure or improve the figure to demonstrate this?

P4. The authors report the performance of methods with the metrics “C-index” and “Brier score”. The authors see an improvement for C-index but the Brier score is similar for most methods. Can the authors please explain or provide some intuitive reasoning for this observation?

P5. In Figure 2, the authors present LSCC recurrence curves.
a) Firstly, can the authors clarify what the y-axis means? Is it better to have a lower value as time progresses?
b) Secondly, in the right-side figure, the blue line (Stage II) curves go below the red line (Stage III) for some time and go above the green line (Stage I) curve towards the end. Can the authors please explain or comment on this behavior?


**Deanonymize Review:**

no

**Final Rating After The Rebuttal:**

4: Weak Accept

**Justification Of The Final Rating:**

The authors have addressed most of the concerns raised and have provided valid justifications or clarifications.
I am happy with the revised draft and have changed the rating accordingly.
I have no further comments.

**Paper Type:**

methodological development

**Questions To Address In The Rebuttal:**

I request the authors to clarify the weakness points as stated above mainly the below ones:
1. The selection of positive and negative samples for the conditional InfoNCE loss that avoids the batch effects [P1,P2].
2. The interpretation of one of the evaluation metric "Brier score" as it has similar values for most methods [P4].
3. More explanation and details required in the caption of Figure 2 [P5].

**Special Issue:**

no

---

### Official Review · Reviewer_LF3h · 2022-01-28

**Confidence:** 5
**Preliminary Rating:** 4
**Recommendation:** Poster

**Summary:**

This paper proposes a conditional self-supervised learning (SSL) approach to predict (2-year) recurrence in Lung Squamous Cell Cancer from H&E stained tissue images. The proposed algorithm first learn WSI representations at the patch-level, and use clustering (with Gaussian Mixture Model) to identify patches with similar histopathological representations. The resulting representations and clusters from self-supervision are then used as features of a Cox proportional hazards model for recurrence prediction at the patient level. Extensive experiments are conducted to demonstrate the effectiveness of the proposed approach in terms of C-index and 2-year Brier score.

**Strengths:**

The paper is clearly written and all methods, experiments, etc. are well described throughout the paper. The authors conducted experiments on two datasets and reported results with correct patient/hospital level splits of the training, testing, and validation sets.

**Weaknesses:**

The authors have reported cross-validation results for two cohorts in the paper. It would have been better if images from one cohort was used for model training and the other cohort was explicitly used for performance assessment - this would have provided an idea about generalizability of the proposed model.

**Deanonymize Review:**

no

**Final Rating After The Rebuttal:**

4: Weak Accept

**Justification Of The Final Rating:**

The authors have successfully addressed the minor issues raised by the reviewer in the previous round. I congratulate the authors for their wonderful work and it would be great to see their paper presented at the conference.

**Paper Type:**

validation/application paper

**Questions To Address In The Rebuttal:**

1. The authors should include a definition of LSCC recurrence in the paper. It is not clear if the authors wanted to predict the risk of 2-year recurrence in this paper? This will help the readers to understand the definition of censoring used in this paper - meaning that all patients who did not recur within 2 years were censored (these patients might have recurred after 2 years of course) of surgery (or diagnosis?) .

2. Since the authors extract multiple patches from WSI of a given patient, it is not clear how the authors combine patch-level results to obtain patient-level risk score  - this detail should be included in the paper.

3. Since the authors have used Cox-Ph model for risk computation, it is not clear how they transform that estimated risk to probability for Brier score calculation. The authors should provide some details of this transformation.

4. The authors used 2-year Brier score as model calibration criteria, but it is established the Brier score is not useful to asses the clinical value of a prediction algorithm (https://pubmed.ncbi.nlm.nih.gov/31093548/). Why not use calibration plots (https://link.springer.com/article/10.1186/s12916-019-1466-7) to assess model calibration instead?

5. Why only Stage was used as a variable to predict recurrence? Several other clinical variables including patient age, race etc. are available in the TCGA datasets - the authors should estimate the recurrence risk using all available clinical variable. There might be a lot of confounding in the reported "Stage" based results. Also, the authors should avoid the term "Stage grading" as Tumor Stage and Tumor Grade are two different things.

6. It is not clear from the paper if the authors used FFFPE or frozen samples from the TCGA/CPTAC data. Also, for each patient there might be multiple WSIs in these datasets - did the authors use all slides for a given patient? These detail should be  included in the paper.

**Special Issue:**

no

---

### Meta-Review · Area_Chair_pb2H · 2022-02-19

**Recommendation:** Accept (Poster)
**Confidence:** 4

**Metareview:**

The paper proposes a simple and effective sampling approach to improve generalisation of survival modelling across different histopathology slides. The authors evaluate their multiple instance learning method on two separate datasets and show good performance.

Advancing self-supervised algorithms in histopathology and medical imaging in general is of great practical importance and is one of the keys to overcoming the data size bottleneck.

Overall the authors have answered reviewers concerns well and the paper reads clearly.

Pros
* simple sampling approach with widespread applicability
* use of open data, multiple datasets

Cons
* Additional benchmarking with various competitive approaches using weak labels and/or transfer learning would be welcome, in particular with/without the sampling technique proposed

---

### Decision · Program_Chairs · 2022-02-28

Accept